# High Fecal Contamination and High Levels of Antibiotic-Resistant Enterobacteriaceae in Water Consumed in the City of Maputo, Mozambique

**DOI:** 10.3390/biology10060558

**Published:** 2021-06-20

**Authors:** Acácio Salamandane, Filipa Vila-Boa, Manuel Malfeito-Ferreira, Luísa Brito

**Affiliations:** Linking Landscape, Environment, Agriculture and Food (LEAF) Research Centre, Instituto Superior de Agronomia, University of Lisbon, 1349-017 Lisbon, Portugal; filipavilaboa@gmail.com (F.V.-B.); lbrito@isa.ulisboa.pt (L.B.)

**Keywords:** water quality, street water, antibiotic resistance, Enterobacteriaceae, beta-lactams, Maputo-Mozambique

## Abstract

**Simple Summary:**

The high number of diarrheal disease cases in developing countries is related to sanitation conditions, consumption of untreated water, and poor individual and collective hygiene. In this study, the microbiological quality of water sold and consumed in the city of Maputo, Mozambique, and the antibiotic resistance profile of Enterobacteriaceae isolated from these samples were evaluated. The results showed the occurrence of microorganisms that indicate fecal contamination with enterococci, fecal coliforms, and *Escherichia coli* above the limit legally allowed for drinking water. The antibiotic resistance profile revealed the existence of antibiotic-resistant bacteria. These results show the need to improve the water supply system in the city of Maputo and to educate the population on hygiene to reduce health risks and promote well-being.

**Abstract:**

In the city of Maputo, Mozambique, food and water are often sold on the streets. Street water is packaged, distributed, and sold not paying attention to good hygienic practices, and its consumption is often associated with the occurrence of diarrheal diseases. Coincidentally, the increase of diarrheal diseases promotes the inappropriate use of antibiotics that might cause the emergence of antibiotic-resistant bacterial strains. In this context, the present study aimed to assess the microbiological quality of water sold on the streets of Maputo, as well as the antibiotic resistance profile of selected Enterobacteriaceae isolates. The 118 water samples analyzed were from street home-bottled water (*n* = 81), municipal water distribution systems (tap water) (*n* = 25), and selected supply wells in several neighborhoods (*n* = 12). The samples were analyzed for total mesophilic microorganisms, fecal enterococci, fecal coliforms, *Escherichia coli*, and *Vibrio* spp. The results showed a high level of fecal contamination in all types of water samples. In home-bottled water, fecal coliforms were found in 88% of the samples, and *E. coli* in 66% of the samples. In tap water, fecal coliforms were found in 64%, and *E. coli* in 28% of the samples. In water from supply wells, fecal coliforms and *E. coli* were found in 83% of the samples. From 33 presumptive *Vibrio* spp. colonies, only three were identified as *V. fluvialis*. The remaining isolates belonged to *Aeromonas* spp. (*n* = 14) and *Klebsiella* spp. (*n* = 16). Of 44 selected Enterobacteriaceae isolates from water samples (28 isolates of *E. coli* and 16 isolates of *Klebsiella* spp.), 45.5% were not susceptible to the beta-lactams ampicillin and imipenem, 43.2% to amoxicillin, and 31.8% to amoxicillin/clavulanic acid. Regarding non-beta-lactam antibiotics, there was a high percentage of isolates with tolerance to tetracycline (52.3%) and azithromycin (31.8%). In conclusion, water in Maputo represents a risk for human health due to its high fecal contamination. This situation is made more serious by the fact that a relatively high percentage of isolates with multidrug resistance (40%) were found among Enterobacteriaceae. The dissemination of these results can raise awareness of the urgent need to reduce water contamination in Maputo and other cities in Mozambique.

## 1. Introduction

In developing countries, waterborne diseases represent the main cause of hospitalization and mortality [1]. Poor water supply systems, weak sanitation and drainage systems, and lack of personal hygiene have been identified as the causes of the increase in the number of outbreaks of various diseases, in particular diarrheal diseases [2,3,4,5]. Most developing countries are unable to provide adequate sanitation to their population, leaving people at risk of water-, sanitation-, and hygiene-related diseases [4,5,6]. These countries are experiencing a shortage of potable water, as improved water sources are limited to urban areas [6,7]. The most reliable source of drinking water is bottled water, which is of good bacteriological quality [1,7]. Furthermore, bottling and the sale of bottled water represents an important milestone in the control of waterborne diseases [8] but it is expensive and, therefore, only accessible to the most privileged sectors of the population [6,7].

However, in many developing countries, in addition to industrial bottled water, home-bottled water is also sold [9]. The microbial quality of this water is not always guaranteed, especially because water is home-bottled in previously discarded plastic bottles [9,10]. The sale of home-bottled water is common in developing countries [7] and is an important source of income in those countries [7,9,10]. In Mozambique, as in most southern African countries, home-bottling involves the reuse of bottles, which are often collected in garbage containers [9]. The process of washing bottles and bottling water for sale does not comply with minimum hygiene criteria [8,9].

Several studies have shown that most bottled water sold in many developing countries is of good physical quality but does not meet the chemical and microbiological quality required for human consumption [1,9,11,12,13,14]. The consumption of these waters contributes to the increase of water-borne diarrheal diseases [3,15,16,17].

In Mozambique, despite the government’s effort to reduce water-borne diseases, there are still many cases of diarrhea caused by water consumption, especially in rainy seasons [3,18,19]. This increases the pressure on the already deficient health system or makes patients turn to informal markets for the sale of antibiotics, which has been increasing in many countries [17,20,21]. However, local studies report that due to the pressure exerted on hospitals, the treatments given to these cases are carried out empirically, which can lead to inappropriate antibiotic consumption [22,23,24,25]. Therefore, this study was carried out to determine the microbiological quality of water sold on the streets of Maputo, Mozambique, and to evaluate the antibiotic resistance profile of Enterobacteriaceae isolates collected from these waters.

## 2. Materials and Methods

### 2.1. Study Area

The city of Maputo, the capital and the most populous Mozambican city, has an estimated area of 347.69 km^2^ and a population density of 3670.6/km^2^ [26]. Maputo has very diverse cultural and ethnic groups resulting from the country’s ethnic diversity and foreign immigrants from central African countries [27,28]. The administrative division of Maputo consists of seven municipal districts [28,29]. This study focused on four of them (Nlhamankulu, KaMaxaquene, KaMavota, and KaMubukwana) because of their abundant street selling activities [29].

### 2.2. Description of the Home-Bottling Water Process for Sale on the Street

The process of home-bottling water for sale on the street begins with the pick-up of plastic bottles discarded in garbage containers or elsewhere. Waste pickers, who sell bottles to street vendors of home-bottled water, usually collect them. The bottles are then washed in a basin, filled with tap water, well water, or another source of water, and frozen in household refrigerators. The source of water for filling the bottles, usually 0.5 L, depends on the source of water normally used by the seller, which may be municipal water or water from a well.

The filling and cooling process usually takes place on the night before the day of sale. Bottles of cooled water (0.5 L) are transported in buckets with lids, such as coolers. In addition to the 0.5 L bottles, 5 L bottles are also packaged with water, frozen, and transported to fill 0.5 L bottles. Consumers drink the water on site and return the bottle to the seller, who immediately refills it for sale to another customer.

### 2.3. Sampling

The 118 water samples analyzed were taken from supply wells in several neighborhoods (*n* = 12), municipal water distribution systems (tap water) (*n* = 25), and home-bottled street water (*n* = 81) (Table 1). The samples were collected from September to October 2019, according to sampling criteria for collecting drinking water [30], immediately placed in a refrigerated (4–8 °C) thermal box, and transported to the National Laboratory of Food Hygiene and Water (Mozambican Ministry of Health). The samples were kept between 4 and 8 °C until microbiological analysis was performed within 6 h after collection.

### 2.4. Microbiological Analysis

The water samples were analyzed for each of the following microbial indicators: total mesophilic microorganisms, fecal enterococci, fecal coliforms, *Escherichia coli*, and *Vibrio* spp.

#### 2.4.1. Enumeration of Total Mesophilic Microorganisms at 22 °C and 37 °C

Enumeration of mesophiles was performed at 22 °C and 37 °C for 72 h, according to ISO standards [31]. Samples (1 mL) were incorporated into Plate Count Agar medium (Biokar Diagnostics, Beauvais, France) and incubated for 72 h at 22 and 37 ± 1 °C. The results were expressed in log CFU/mL.

#### 2.4.2. Enumeration of Fecal Enterococci, Fecal Coliforms, and *Escherichia coli*

The enumeration of fecal enterococci, fecal coliforms, and *E. coli* was performed after filtration of 100 mL of water through a membrane with a 0.22 µm pore size (Cellulose Nitrate Filter, Sartorius, Germany). The results were expressed in log CFU/100 mL. Typical colonies of fecal enterococci (red, maroon, or pink) were counted on Slanetz and Bartley Agar medium (Biokar Diagnostics) after 24 h of incubation at 42 ± 2 °C, according to ISO standards [31]. The typical colonies of fecal coliforms (dark blue/violet) and *E. coli* (pink to red) were counted on Chromogenic Coliform Agar (CCA) medium (Biokar Diagnostics) after 24 h of incubation at 44 ± 1 °C, according to ISO standards [32].

#### 2.4.3. Enumeration of Presumptive *Vibrio* spp.

Enumeration of *Vibrio* spp. was conducted by filtration of 1 L water with a 0.22 µm filtration membrane (Cellulose Nitrate Filter) and incubation on *Vibrio* Chromogenic Agar (Candalab, Madrid, Spain) for 24 h at 36 ± 1 °C. This medium is recommended for the selective isolation and differentiation of *Vibrio* spp. based on colony color: *V. cholerae* (pink-rose), *V. alginolyticus* (colorless), *V. parahemolyticus* (green-blue), and *V. vulnificus* (pink-rose).

#### 2.4.4. Biochemical Characterization and Molecular Identification of Bacteria

Biochemical tests were carried out to further characterize the colonies of presumptive *E. coli* and *Vibrio* spp. from the respective chromogenic media, namely, Gram staining, catalase and oxidase tests, and growth with NaCl (6% *m*/*v*) in the medium (*Vibrio* spp. only).

Subsequently, molecular identification was performed, targeting the 16S rRNA gene (401 bp) with the AB035924 forward primer (5′ CCC CCT GGA CGA AGA CTG A-3′) and AB035924 reverse primer (5′-ACC GCT GGC AAC AAA GGA T-3′) [33] or with the Bac27F forward primer (5-AGAGTTTGGATCMTGGCTCAG-3) and Univ1492R universal reverse primer (5-CGGTTACCTTGTTACGACTT-3) [34]. The amplified products were sequenced by STAB VIDA (Caparica, Portugal), and the resulting sequences were compared in the BLASTN gene bank [35]. The accepted percentage of identity was greater than 90% similarity [36,37].

### 2.5. Antibiotic Susceptibility Profile

The antibiotic susceptibility profile was evaluated for 44 isolates of Enterobacteriaceae (28 *E. coli* and 16 *Klebsiella* spp. isolates) using the disc diffusion method on Mueller–Hinton (MH) agar plates (Biokar Diagnostics, Beauvais, France) with antibiotic disc (Liofilchem, Roseto degli Abruzzi, Italy) according to the Clinical Laboratory Standards Institute (CLSI) [38]. Isolated colonies grown on Trypto–Casein–Soy agar (Biokar Diagnostics) for 22 ± 2 h at 37 °C were suspended in sterile saline until the turbidity was equivalent to the Mc Farlands 0.5 standard. The resulting bacterial suspensions were used to inoculate MH plates. After antibiotic disc deposition, the plates were incubated for 18 ± 2 h at 37 °C. Fifteen antibiotics were used: amoxicillin (AMX) 10 mg; amoxicillin/clavulanic acid (AMC) 30:10 mg; ceftazidime (CAZ) 30 mg; imipenem (IPM) 10 mg; cefpirome (CPO) 30 mg; aztreonam (ATM) 30 mg; cefoxitin (FOX) 30 mg; ampicillin (AMP) 10 mg; cefotaxime (CTX) 30 mg; chloramphenicol (CHL) 30 mg; tetracycline (TET) 30 mg; gentamycin (GEN) 10 mg; trimethoprim/sulfamethoxazole (SXT) 1:19 mg; azithromycin (AZM) 15 mg; ciprofloxacin (CIP) 5 mg. In each 90 mm plate, five different antibiotic discs were applied, and two replicates were used for each antibiotic. The ranges of the diameter of each antibiotic disc for analysis according to CLSI [38] are in Appendix A.

### 2.6. Data Interpretation

For each type of water sample and indicator microorganism, the average values of log CFU/mL or log CFU/100 mL and the respective standard deviations from duplicate plates were calculated. For the interpretation of the results, the legislative decree for the microbiological quality of drinking water issued by the Ministry of Health of Mozambique [39], the European Council Directive 98/83/CE [40], and the recommendations of WHO [41] were used. For the evaluation of antibiotic resistance, the diameters of the inhibition halos (mm) were measured and compared to those described by the CLSI [38]. The isolates were considered non-susceptible to a certain antibiotic when they showed intermediate or full resistance to that antibiotic. Multidrug resistance was considered as resistance to more than two unrelated antimicrobial agents.

## 3. Results

The water samples were analyzed for different microbial indicators, according to different guidelines, directives, and recommendations for the microbiological quality of drinking water. The identification of selected colonies of presumptive *E. coli* and *Vibrio* spp. collected from the chromogenic media were confirmed by sequencing within the 16S rRNA gene with an accepted percentage of identity greater than 90% similarity [36,37,42,43]. The 28 isolates selected from CCA plates were confirmed as *E. coli* (Table 2). However, of the 33 isolates selected from the Vibrio Chromogenic Agar plates presumed to be *Vibrio* spp., only three isolates (from home-bottled street water) were confirmed as *V. fluvialis* (Table 2). The other isolates were identified mainly as belonging to *Aeromonas* spp. (14) (42.4%) and *Klebsiella* spp. (16) (48.5%) (Table 2). Ten *Aeromonas* spp. isolates were from home-bottled street water, and four from well water.

### 3.1. Microbiological Quality of Water

#### 3.1.1. Piped Water (Tap Water)

The tap water samples showed a level of microbiological contamination higher than that recommended for human consumption (Table 3). Of the samples, 92% and 96% contained aerobic mesophiles at 22 and 37 °C, respectively, that were greater than 2 log CFU/mL and classified as unsatisfactory according to the European Council Directive 98/83/CE [24]. Regarding fecal enterococci, 68% of the samples presented values greater than 1.9 log CFU/100 mL (Table 3). Fecal coliforms and *E. coli* were found in 64 and 28% of the water samples, respectively (Table 3), and were classified as unsatisfactory according to the European Council Directive 98/83/CE [24], WHO guidelines [44], and Mozambican directives [39].

#### 3.1.2. Water from Supply Wells

The most contaminated water samples were those from a well supply. Aerobic mesophiles and fecal enterococci were found in the 12 water samples (100%) (Table 4). Fecal coliforms and *E. coli* were found in 83% of the samples. *Aeromonas* spp. (33.3%) and *Klebsiella* spp. (25%) (Table 2) were also found in all samples.

#### 3.1.3. Home-Bottled Street Water

In the 81 samples of home-bottled street water, mesophiles at 22 and 37 °C were found in all samples, with contamination levels above 2.3 log UFC/mL (Table 5). Fecal enterococci were found in 65% of the water samples, with a mean value higher than 1.5 log CFU/100 mL. Fecal coliforms and *E. coli* were found in 54 and 31% of the samples, respectively.

### 3.2. Antibiotic Resistance Profile

Evaluation of the antimicrobial susceptibility of 44 isolates of Enterobacteriaceae collected from the water samples (28 *E. coli* and 16 *Klebsiella* spp.) to 15 antimicrobial agents (Table 6) showed the presence of isolates tolerant to the ß-lactams AMP and IPM (45.5%), AMX (43.2%), and AMC (31.8%). For non-beta-lactam antibiotics, there was a high percentage of isolates tolerant to TET (52.3%) and SXT (31.8%) (Table 6).

A higher percentage of *Klebsiella* isolates were tolerant to beta-lactams AMC (62.5%), AMX (56.3%), and CTX (35.7%), compared to *E. coli* isolates (14.3, 35.7, and 10.7%, respectively; Appendix A). However, for FOX and IPM, the percentage of resistant isolates was very similar for the two genera. For non-beta-lactams, more *E. coli* isolates were resistant to SXT (39.3%) and CHL (21.4%) than *Klebsiella* spp. (18.8 and 6.3%, respectively). For TET, a similar profile of susceptibility was observed for the two genera (Appendix A). Moreover, 30 of the 44 Enterobacteriaceae isolates (68.2%) were resistant to at least one antibiotic (Table 7). There was a high prevalence (48.6%) of Enterobacteriaceae isolates with multidrug resistance (Table 7); this occurred more frequently in *E. coli* (67%) than in *Klebsiella* spp. (33%) (Appendix A).

## 4. Discussion

### 4.1. Quality of the Water Collected in Maputo

Despite the risk to public health, the reuse of disposable containers for food products is frequent in many low-income countries [9]. The lack of health education and supervision has contributed to the increase in the reuse of containers for packaging food and beverages, especially water [9]. In the case of Maputo, this is a business opportunity for countless families, concerning water and street food, or water containers [10]. The high demand for low-cost bottled water has contributed to the growth of the number of sellers [10].

In general, the occurrence of fecal contamination was high and may represent a risk to the health of water consumers in the city of Maputo. For home-bottled street water, a high level of contamination was found for all microbiological indicators. In Nnewi, Nigeria, 42% of samples of home-bottled and non-bottled street water were contaminated with *E. coli* [1]. Studies carried out in India reported that the quality of home-bottled street water was poor, with high levels of mesophiles [10] and, in some cases, suspended particles in the bottles [10,14]. At a first glance, the reuse of bottles could be interpreted as the main cause of this contamination. However, despite the smaller number of piped water (*n* = 25) and well water samples (*n* = 12) evaluated in this study in comparison with home-bottled water (*n* = 81) (Table 1), the analysis of piped (tap) and supply well water collected in four neighborhoods in the city of Maputo showed high levels of mesophiles, fecal enterococci, and coliforms.

The presence of indicators of fecal contamination, such as fecal enterococci, fecal coliforms, and *E. coli*, in the water collected from supply wells suggests contamination of the water table by septic tanks (latrines) that commonly exist in almost all peripheral neighborhoods. Consequently, in addition to the reuse of bottles, the poor microbiological quality of the water used for bottling also contributed to the high level of contamination of home-bottled street water. Furthermore, the quality of water depends on the depth of the water table and the presence of contamination sources such as untreated sewage and latrines [10]. Studies conducted in Nairobi [44] and Malawi [45] also showed high levels of *E. coli*, total coliforms, and fecal coliforms (>2 log CFU/100 mL) in water from wells. In South Africa (Eastern Cape Province) [46] and Mohale Basin, Lesotho [47], the poor quality of domestic water sources in selected communities was also reported. Therefore, water disinfection should be included in the water treatment process [48].

Failures in the water treatment system have been reported in several cities in Mozambique, including Maputo [49]. In the city of Maputo, the water supply pipeline has not been properly maintained. It is very old and broken in several sections [50]. Whenever it rains and there are floods, the water supply network is submerged, which causes contamination, since the water comes from open drainage ditches [49,51]. Studies in several southern African countries showed that tap water supplied to consumers was not of good microbiological quality [52].

This work was carried out in Maputo, Mozambique, during the dry season (September to October). If it had been performed during the rainy season (December to April), the levels of contamination, particularly, with enteric bacteria, would have been higher due to the floods that occur cyclically in various regions of Mozambique [3,28]. In addition, the presence of *V. cholera* would be expected [53].

Although several authors have confirmed the efficiency of *Vibrio* Chromogenic Agar medium [54,55,56,57], it was not confirmed in the present study. Of the 33 presumptive *Vibrio* isolates, only three were confirmed as *Vibrio* spp. and non-pathogenic *V. fluvialis*. The other 30 isolates were identified as *Aeromonas* spp. and *Klebsiella* spp. Perry [55] stated that the medium is efficient for the detection of species of pathogenic *Vibrio*, such as *V. cholerae* and *V. parahaemolyticus*, although it was less effective (50%) in differentiating these two species compared to other media, such as Thiosulfate Citrate Bile Salts Sucrose (TCBS) Agar. Canizalez-Roman et al., 2011 [54] also observed the growth of presumptive *Vibrio* colonies on *Vibrio* Chromogenic Agar that were identified as *Listeria monocytogenes*, *Aeromonas caviae*, *Aeromonas* spp., and *Pseudomonas* spp.

### 4.2. Antibiotic Resistance

In this study, a high prevalence of resistance to beta-lactams was found. AMX and AMP were the beta-lactams with the highest percentage of resistant isolates (38.6 and 38.4%, respectively; Table 6). In the group of non-beta-lactam antibiotics, TET and SXT had the highest percentage of resistant isolates (52.3 and 31.8%, respectively; Table 6). These results reflect the existence of selective pressures due to the use of antimicrobials in the prevention and treatment of human diseases [58]. To our knowledge, this is the first study to investigate the occurrence of antibiotic resistance in isolates of Enterobacteriaceae collected in water intended for human consumption in Mozambique. Some studies with isolates from Mozambican hospital settings [22,59,60] have shown a high prevalence of resistance to β-lactams and non-β-lactams. Chirindze et al. [60] reported a high incidence of resistance for STX and TET (greater than 60%) in clinical isolates of *E. coli* and *Klesibiella* spp.

The high incidence of resistance to IPM in this study was due to the high percentage (65%) of intermediate susceptibility (non-full resistance). The high percentage of individuals with intermediate susceptibility may be a relevant indicator in the treatment of future clinical cases [61]. In another study with clinical isolates performed in Beira, Mozambique [59], a high prevalence (63 to 100%) of resistance to the β-lactams AMX, CPO, AMC, and CAZ as well as to the non-β-lactams STX, CHL, and TET (80–100%) was reported. However, this work also reported no resistance to IPM. In the study of Mandomando et al. (2010) [22] carried out in Maputo, CHL and AMP showed the least effectiveness in controlling clinical isolates of Enterobacteriaceae, with an incidence of resistance of 60% and 80%, respectively.

There was a prevalence of 48.6% of isolates with multidrug resistance (Table 7), which may be related to the high frequency of antibiotics use, sometimes without medical prescription, that has been favored by the existence of informal markets for the sale of antibiotics, as mentioned by several authors [20,21,62,63]. Another factor that can contribute to this high resistance is the practice of taking antibiotics in incomplete doses [21].

## 5. Conclusions

The results of this study showed high fecal contamination of home-bottled water sold on the streets of Maputo, Mozambique. Therefore, this water is unfit for human consumption. Since most consumers of this water are people of low income and generally with low education, it is important to introduce awareness campaigns about the danger of consuming untreated water. Promoting a short course on methods of water treatment and preparation and handling of ready-to-eat foods can reduce the risk of outbreaks of waterborne and foodborne illnesses. Likewise, the water supplied by the municipal supply system is also of poor microbiological quality. As it is considered drinking water, the population may be consuming this water without any additional treatment, which represents a risk to public health. Furthermore, high levels of antibiotic-resistant and multidrug-resistant Enterobacteriaceae were observed in the water samples. These results highlight the importance of assessing antibiotic resistance when evaluating the potential risk associated with drinking this water. The high percentage of antibiotic resistance in water isolates may be related to the occurrence of environmental strains with antibiotic resistance. However, the existence of cases of antibiotic consumption without a prescription in the local community may be contributing to the increase in resistance to antibiotics.

## Figures and Tables

**Table 1 biology-10-00558-t001:** Number and type of water samples collected in Maputo municipal districts.

Sample Type	Maputo Municipal District				Total
	Nlhamankulu	KaMaxaquene	KaMubukwana	KaMavota	
Tap water	5	7	7	6	25
Home-bottled street water	20	20	20	21	81
Water from supply wells	-	4	4	4	12

**Table 2 biology-10-00558-t002:** Source, number, and identification of selected isolates by 16S rRNA gene sequencing.

Microbial Group	Species	Home Bottled Water	Tap Water	Supply Well Water	Total
Enterobacteriaceae	*Escherichia coli **	20	2	6	28
	*Klebsiella oxytoca*	4	-	1	6
	*Klebsiella aerogenes*	7	-	4	11
	Total	31	2	11	44
non-Enterobacteriaceae	*Aeromonas hydrophila*	3	-	2	5
	*Aeromonas veronii*	4	-	1	5
	*Aeromonas caviae*	3	-	1	4
	*Vibrio fluvialis*	3	-	-	3
	Total	13	-	4	17

* *E. coli* isolates were collected from CCA plates. The other isolates were collected from *Vibrio* chromogenic Agar plates as presumptive *Vibrio* spp.

**Table 3 biology-10-00558-t003:** Evaluation of piped water (tap water) in four municipal districts of Maputo, according to each microbial indicator for drinking water. Number of samples with contamination higher than the limits and total percentage (%) are indicated.

Microbial Indicator **	Piped Water Distribution Area				Total	Limits (WHO/EU/DM *)
	Nlhamankulu	KaMaxaquene	KaMubukwana	KaMavota		
Mesophiles at 22 °C	2.47 ± 0.41 5/5	2.37 ± 0.09 7/7	2.37 ± 1.09 5/7	2.11 ± 0.42 5/6	2.34 ± 0.7 22/25 (92%)	2 log/mL
Mesophiles at 37 °C	2.28 ± 0.09 5/5	2.19 ± 0.26 7/7	2.23 ± 1.04 5/7	2.29 ± 0.5 5/6	2.23 ± 0.67 24/25 (96%)	1.3 log/mL
Fecal *enterococci*	1.97 ± 0.98 3/5	1.77 ± 0.26 6/7	1.97 ± 0.88 5/7	0.99 ± 1.01 3/6	1.97 ± 0.9 17/25 (68%)	Absence in 100 mL
Fecal coliforms	1.94 ± 0.96 3/5	1.76 ± 0.64 6/7	1.79 ± 0.95 4/7	0.9 ± 0.96 3/6	1.79 ± 0.9 16/25 (64%)	Absence in 100 mL
*E. coli*	3/5 (0.95 ± 0.63)	Absent	4/7 (0.88 ± 0.62)	Absent	0 ± 0.56 7/25 (28%)	Absence in 100 mL

* The legislative decree for the microbiological quality of drinking water issued by the Ministry of Health of Mozambique (Ministerial Diploma 180/2004) [39]. ** Average values ± standard deviation of log CFU/mL for mesophiles at 22 °C and 37 °C or log CFU/100 mL for fecal enterococci, fecal coliforms, and *E. coli*).

**Table 4 biology-10-00558-t004:** Evaluation of water from supply wells in three municipal districts of Maputo, according to each microbial indicator for drinking water. Number of samples with contamination higher than the limits and total percentage (%) are indicated.

Microbial Indicator **	Supply Well Area			Total	Limits (WHO/EU/DM *)
	KaMaxaquene	KaMubukwana	KaMavota		
Mesophiles at 22 °C	2.44 ± 0.04 4/4	2.47 ± 0.03 4/4	2.5 ± 0.02 4/4	2.49 ± 0.04 12/12 (100%)	2 log/mL
Mesophiles at 37 °C	2.08 ± 0.07 4/4	2.2 ± 0.12 4/4	2.12 ± 0.15 4/4	2.11 ± 0.12 12/12 (100%)	1.3 log/mL
Fecal enterococci	1.85 ± 0.06 4/4	1.85 ± 0.19 4/4	1.89 ± 0.09 4/4	1.85 ± 0.13 12/12 (100%)	Absence in 100 mL
Fecal coliforms	1.95 ± 0.08 4/4	1.91 ± 0.07 4/4	1.82 ± 0.16 4/4	1.91 ± 0.12 10/12 (83%)	Absence in 100 mL
*E. coli*	1.31 ± 0.59 3/4	1.18 ± 0.053 3/4	1.32 ± 0.1 3/4	1.27 ± 0.49 10/12(83%)	Absence in 100 mL

* The legislative decree for the microbiological quality of drinking water issued by the Ministry of Health of Mozambique (Ministerial Diploma 180/2004) [39]. ** Average values ± standard deviation of log CFU/mL for mesophiles at 22 °C and 37 °C or log CFU/100 mL for fecal enterococci, fecal coliforms, and *E. coli*).

**Table 5 biology-10-00558-t005:** Evaluation of home-bottled street water sold in four municipal districts of Maputo, according to each microbial indicator for drinking water. Number of samples with contamination higher than the limits and total percentage (%) are indicated.

Microbial Indicator **	Home-Bottled Street Water Sales Area				Total	Limits (WHO/ EU/DM *)
	Nlhamankulu	KaMaxaquene	KaMubukwana	KaMavota		
Mesophiles at 22 °C	2.47 ± 0.1 20/20	2.36 ± 0.13 20/20	2.39 ± 0.08 20/20	2.35 ± 0.25 20/20	2.41 ± 0.16 81/81 (100%)	2 log/mL
Mesophiles at 37 °C	2.12 ± 0.22 20/20	2.15 ± 0.09 20/20	1.96 ± 0.3 20/20	2.15 ± 0.38 21/21	2.10 ± 0.29 81/81 (100%)	1.3 log/mL
Fecal *enteroccoci*	2.1 ± 0.8 17/20	1.92 ± 0.86 15/20	1.95 ± 0.67 16/20	1.85 ± 0.76 16/21	1.51 ± 0.86 65/81 (80.2%)	Absence in 100 mL
Fecal coliforms	2.14 ± 0.29 20/20	2.08 ± 0.46 19/20	2.03 ± 0.81 16/20	2.12 ± 0.87 14/21	2.07 ± 0.69 72/81 (88.9%)	Absence in 100 mL
*E. coli*	2.82 ± 0.83 17/20	1.49 ± 0.79 13/20	1.53 ± 0.93 11/20	1.3 ± 0.78 12/21	1.94 ± 0.82 54/81 (66.7%)	Absence in 100 mL

* The legislative decree for the microbiological quality of drinking water issued by the Ministry of Health of Mozambique (Ministerial Diploma 180/2004) [39]. ** Average values ± standard deviation of log CFU/mL for Mesophiles at 22 °C and 37 °C or log CFU/100 mL for fecal enterococci, fecal coliforms, and *E. coli*).

**Table 6 biology-10-00558-t006:** Antibiotic resistance profiles of 44 isolates of *Enterobacteriaceae* (28 isolates of *E. coli* and 16 isolates of *Klebsiella* spp.).

Type of Antibiotic	Antibiotic *	Antibiotic Susceptibility Pattern Number of Isolates (Percentage)			
		Susceptible	Non-Susceptible		
			Intermedium	Resistant	Total
Beta-lactam	AMX (10 μg)	25 (56.8%)	2 (4.5%)	17 (38.6%)	19 (43.2%)
	AMC (20/10 μg)	30 (68.2%)	4 (9%)	10 (22.7%)	14 (31.8%)
	CAZ (30 μg)	39 (88.6%)	1 (2.3%)	4 (9%)	5 (11.4%)
	IPM (10 μg)	24 (54.4%)	13 (29.5%)	7 (15.9%)	20 (45.5%)
	CPO (30 μg)	42 (95.5%)	#	2 (4.5%)	2 (4.5%)
	ATM (30 μg)	39 (88.6%)	0	5 (11.4%)	5 (11.4%)
	FOX (30 μg)	33 (75%)	2 (4.5%)	9 (20.5%)	11 (23.4%)
	AMP (10 μg)	24 (54.4%)	4 (9%)	16 (36.4%)	20 (45.5%)
	CTX (30 μg)	34 (77.3%)	5 (11.4%)	5 (11.4%)	10 (22.7%)
Non-beta-lactam	CHL (30 μg)	37 (84.1%)	4 (9%)	3 (6.8%)	7 (15.9%)
	TET (30 μg)	21 (47.7%)	7 (15.9%)	16 (36.4%)	23 (52.3%)
	GEN (10 μg)	37 (84.1%)	3 (6.8%)	4 (9%)	7 (15.9%)
	SXT (23.75/1.25 μg)	30 (68.2%)	2 (4.5%)	12 (27.3%)	14 (31.8%)
	AZM (15 μg)	35 (79.4%)	#	9 (20.5%)	9 (20.5%)
	CIP (5μg)	42 (95.5%)	1 (2.3%)	1 (2.3%)	2 (4.5%)

* Amoxicillin (AMX) 10 mg; amoxicillin/clavulanic acid (AMC) 30:10 mg; ceftazidime (CAZ) 30 mg; imipenem (IPM) 10 mg; cefpirome (CPO) 30 mg; aztreonam (ATM) 30 mg; cefoxitin (FOX) 30 mg; ampicillin (AMP) 10 mg; cefotaxime (CTX) 30 mg; chloramphenicol (CHL) 30 mg; tetracycline (TET) 30 mg, gentamycin (GEN) 10 mg, trimethoprim/sulfamethoxazole (SXT) 1:19 mg, azithromycin (AZM) 15 mg; ciprofloxacin (CIP) 5 mg. # The guidelines of CCLS [22] do not show the intermedium range for this antibiotic.

**Table 7 biology-10-00558-t007:** Prevalence of multidrug resistance in a total of 30 non-susceptible *Enterobacteriaceae*.

Type of Resistance	Group of Antibiotics *	Number of Isolates (Percentage)
Multi resistant	AMX, AMC, CAZ, AMP, CHL, SXT	1 (2.9%)
	IPM, CHL, TET, AZM	1 (2.9%)
	AMX, FOX, AMP, TET, SXT	1 (2.9%)
	IPM, GEN, SXT	1 (2.9%)
	TET, GEN, SXT	1 (2.9%)
	AMX, AMC, IPM, FOX, AMP, GEN, SXT, AMZ	1 (2.9%)
	AMC, CAZ, IMP, CPO, ATM, FOX, CTX, CHL, TET, GEN, AZM	1 (2.9%)
	AMX, AMP, CHL, SXT	1 (2.9%)
	AMX, AMC, IMP, AMP, TET, SXT	1 (2.9%)
	CAZ, IPM, CPO, ATM, FOX, CTX, TET, GEN, SXT, AZM, CIP	1 (2.9%)
	AMX, AMP, TET, SXT	1 (2.9%)
	AMX, AMC, IPM, FOX, AMP, CTX, TET, AZM	1 (2.9%)
	AMX, AMC, IPM, FOX, AMP, TET, GEN, SXT	1 (2.9%)
	AMX, CAZ, IPM, FOX, AMP, CHL, TET, AZM	1 (2.9%)
	AMX, AMC, IPM, FOX, AMP, CTX, TET, GEN	1 (2.9%)
	AMX, AMC, AMP, TET AZM	1 (2.9%)
	AMX, AMC, IPM, ATM, FOX, AMP, CTX, TET GEN, AZM	1 (2.9%)
Total		17 (48.6%)
Non-multi resistant	CTX, TET	1 (2.9%)
	IPM, AMP, CTX, TET	1 (2.9%)
	IPM, AMP, TET	1 (2.9%)
	AMX, CAZ, IMP, ATM, FOX, AMP, CTX, CHL	1 (2.9%)
	AMX, AMC, AMP, TET	1 (2.9%)
	TET, SXT	1 (2.9%)
	SXT	2 (5.7%)
	AMX, AMP, TET	1 (2.9%)
	TET, SXT	1 (2.9%)
	AMP, TET	1 (2.9%)
	IMP	1 (2.9%)
	CHL	1 (2.9%)
	AMX, AMC, IPM, FOX, AMP, TET	1 (2.9%)
	AMX, AMC, AMP, CTX, TET	1 (2.9%)
	ATM, CTX	1 (2.9%)
	AMX, AMC, IPM, FOX, AMP	1 (2.9%)
	AMX, AMC, IPM, TET	1 (2.9%)
Total		18 (51.4%)

* Amoxicillin (AMX) 10 mg; amoxicillin/clavulanic acid (AMC) 30:10 mg; ceftazidime (CAZ) 30 mg; imipenem (IPM) 10 mg; cefpirome (CPO) 30 mg; aztreonam (ATM) 30 mg; cefoxitin (FOX) 30 mg; ampicillin (AMP) 10 mg; cefotaxime (CTX) 30 mg; chloramphenicol (CHL) 30 mg; tetracycline (TET) 30 mg, gentamycin (GEN) 10 mg, trimethoprim/sulfamethoxazole (SXT) 1:19 mg, azithromycin (AZM) 15 mg; ciprofloxacin (CIP) 5 mg.

## Data Availability

The sequences resulting from this study were deposited in the GenBank, with the access code: SUB9814615 to SUB9814630.

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
