# Peer review of "High Fecal Contamination and High Levels of Antibiotic-Resistant Enterobacteriaceae in Water Consumed in the City of Maputo, Mozambique"

_biology, 2021, doi:10.3390/biology10060558_

Round 1

Reviewer 1 Report

The authors investigated the microbial contamination level of various water samples obtained from home-bottled water sold in street vendors, tap water, and supply wells in Maputo, Mozambique; and investigated the antibiotic resistance profile of Enterobacteriaceae isolates originated from these water samples. Fecal contamination was observed at a high frequency among the water samples and antibiotic resistance was also often encountered among the Enterobacteriaceae isolates tested in this study. This study provides evidence that unsanitary water represents a serious risk of public health in this study, which could be valid in other developing countries; however, the current manuscript still needs to be improved in various aspects as shown below.

  1. The authors did not explain why they examined the antibiotic resistance profile of the bacteria isolated from the water samples. In lines 13 and 14 in Abstract, this was briefly mentioned but the cause-effect relationship of the unsanitary water sold on the streets and the inappropriate use of antibiotics is not clear without any supporting information for this claim in the manuscript.
  2. Discussion needs to be further streamlined and re-organized as suggested in the specific comments.
  3. The Conclusion section is simply the summary of the data presented in this study and did not provide any meaningful insights. I recommend merging Conclusion with Discussion.
  4. How many disks per antibiotic were placed on one plate?
  5. Cite tables more frequently.
  6. For tables, specify in the legend which unit (log CFU/ml or log CFU/100 ml) was used for each indicator.
  7. Unitalicize "enterococci" and "Enterobacteriaceae".

Specific comments:

Abstract

          Line 11: Show the country where Maputo is located.

          Line 12: I do not think that the acronym "GHP" is necessary here since it was used only once in the abstract.

          Line 16: Clearly mention that these Enterobacteriaceae isolates originated from water samples.

          Lines 19 and 20: Show the full name for "E. coli".

          Line 21: Change "home bottled water" to "home-bottled water" throughout the manuscript. Revise other similar instances accordingly.

          Lines 21-23: Nothing was mentioned pertaining to the water samples from supply wells. Why?

          Line 25: Change "28 of E. coli" to "28 isolates of E. coli".

          Lines 26 and 27: Show the full names for antibiotics, not acronyms.

          Line 26: Change "non b-lactam" to "non-b-lactam".

          Line 27: "higher percentage" than what? Also, change "were to" to "was to".

          Line 32: Be more specific regarding "similar environmental conditions".

Introduction

          Lines 47-49 and 51-53: Provide references for these statements.

          Line 53: Revise "informally bottled or packaged" for clarity.

          Lines 59 and 60: I would suggest changing "have the chemical and microbiological quality" to "meet the chemical and microbiological quality".

          Lines 61 and 62: Show the country where Maputo resides.

Materials and Methods

          Line 68: Re-write "gather national and foreign immigrants" for clarity. What does "national immigrants" mean?

          Line 70: Change "will focus on" to "focused on".

          Line 71: Change "KaMubukwana)," to "and KaMubukwana),". Re-write "the main street selling activities take place".

          Lines 75 and 76: Re-write "bottled water at home" and "do collection" to improve clarity.

          Line 76: Remove "in water" in "washed in water" since "in water" seems to be redundant this sentence.

          Lines 78 and 79: Re-write "street vendors residential area".

          Line 79: What does "used for own consumption" mean?

          Line 89: Re-write "65 street vendor women and 16 street vendor children". In fact, I do not think that this demographic information is necessary.

          Line 102: Show the manufacturer or composition of Ringer Solution.

          Line 102: Change "ISO Standard" to "ISO standard" throughout the manuscript.

          Lines 107, 117, 120, and 171: Italicize "Escherichia coli" and "Vibrio". Revise similar instance accordingly.

          Line 110: Replace the comma next to "Cellulose Nitrate Filter" with a semi-colon.

          Line 112: "Beauvais, France" was already shown. Remove this.

          Line 118: Change "tested" to "conducted".

          Line 121: Change "Vibrio species" to "Vibrio spp.".

          Lines 122 and 123: Change "Vibrio" to "V. ".

          Line 126: Change ", namely:" to ", namely,".

          Line 127: Re-write "catalase, oxidase and the presence of NaCl (6% m/v) in the growth medium" to improve clarity.

          Line 134: Change "(STAB VIDA, Caparica, Portugal)" to "in STAB VIDA (Caparica, Portugal)".

          Line 135: Re-write "in BLASTN gene bank" and provide reference(s) for BLAST.

          Line 135: Change "percentage of identity accepted" to "accepted percentage of identity".

          Line 138: Change "evaluated in" to "evaluated for".

          Lines 139, 224 and 228: Change "(28 E. coli and 16 Klebsiella spp.)" to "(28 E. coli and 16 Klebsiella spp. isolates)".

          Lines 141 and 142: Show the acronym for "Clinical Laboratory Standards Institute" and use the acronym afterwards. In other words, change "Clinical Laboratory Standards Institute (CLSI)" to "CLSI" in lines 160 and 161.

          Line 142: Change "grown onto Trypto- Casein-Soy agar" to "grown on Trypto-Casein-Soy agar".

          Lines 146-151, 229-233 and 247-251: Change capital letters in the names of the antibiotics to lower case. What does "1:19 mg" mean? Add "and" before "Ciprofloxacin".

          Lines 151 and 152: Re-write "The respective ... Table S.1." for clarity.

          Line 162: Change "when had" to "when they showed".

Results

          Lines 165-167: This statement needs to be more specific.

          Line 169: Re-write "sequencing within ... than 90%".

          Line 175: change "four were from well water" to "four from well water".

          Lines 181 and 182: I do not think that this statement is necessary, which is simply a table legend.

          Lines 182-184: Re-write this statement to improve clarity.

          Line 193: Include "Ministerial Diploma 180/2004" in the references.

          Line 195: Re-write the statement "Supply wells ... water".

          Lines 195-197: Show both the number of the positive samples and percentage.

          Lines 205 and 206: I would recommend emphasizing that mesophiles were found in all the bottled water samples. Re-write "bottled street water at home" to improve clarity.

          Line 206: Change "Fecal enterococcus" to "Fecal enterococci".

          Line 208: Change "founded" to "found".

          Lines 214-221: This seems to be more suited to the Discussion. In line 218, it is not clear what "validated" means in this statement. I suggest revising it.

          Lines 226 and 227: "higher percentage" than what?

          Lines 235-237: Cite "Table S.2".

          Line 242: Revise "showed a resistance profile to at least one antibiotic" for clarity.

          Line 243: Change “Klesebiella spp.” to “in Klebsiella spp.”.

          Line 246: Change "non susceptible" to "non-susceptible".

          Line 252: Change "signed" to "marked".

Discussion

          Lines 255-282: This paragraph is too lengthy and I recommending dividing this into shorter paragraphs, which could start from lines 261 and 272.

          Line 260: Revise "evolution" for clarity.

          Line 262: Revise "At first" for clarity.

          Line 266: Change "as fecal enterococci" to "such as fecal enterococci". Unitalicize "and" in "and E. coli".

          Lines 275-278: Use the past tense. Provide references for the statement in lines 276-278.

          Lines 279-282: I suggest moving this statement to line 272 since it is related to the water from supply wells.

          Lines 283-286: This paragraph could be moved to line 262.

          Lines 287-292: This paragraph seems to be redundant with lines 262-268 to some degree and could be merged with the section in lines 262-268. Also, I suggest adding "n=" before the number of samples.

          Line 288: Remove the comma in "in this study," and add one after "bottled water (81)".

          Line 293: Change "Maputo-Mozambique" to "Maputo, Mozambique, ".

          Lines 294-296: Provide references for this statement.

          Line 297: Change "Vibrio cholera" to "V. cholera".

          Line 299: Revise "have the expected results" for clarity.

          Lines 299, 304 and 307: Change "vibrio" to "Vibrio" and italicize it.

          Lines 303 and 307: Remove the publication year right next to the authors throughout the manuscript.

          Lines 311-332: This paragraph is too lengthy and needs to be divided.

          Lines 311-316: The statements "None ... respectively" seem to be more suited to the Results section. I recommend summarizing key findings more succinctly.

          Lines 317 and 318: Change "As far as we know" to "To our knowledge".

          Line 320: "However" does not match the context.

          Lines 322 and 323: Revise "reported for ... TET" for clarity. Does "greater than 60% resistance" refer to the prevalence or intensity? This question also applies to "higher resistance to IPM" in line 323, "with resistance of 60 and 80%" in line 331, and "this high resistance" in line 336.

          Line 324: Change "percentages (65%)" to "percentage (65%)".

          Line 330: I do not think that "Mozambique" is necessary here.

          Lines 333 and 334: Revise "local habits ... authors" for clarity.  

Author Response

Dear reviewer, thank you for all your comments and contributions to the improvement of our manuscript. The highlighted of the answers were posted in yellow.

Reviewer 2 Report

Dear authors,

Manuscript "High fecal contamination and high levels of antibiotic resistant Enterobacteriaceae in the water consumed in the city of Maputo, Mozambique" and authored by Acácio Salamandane, Filipa Vila Boa, Manuel Malfeito-Ferreira and Luísa Brito targets a hot field. I am quite sure that results will be highly interesting for the readers. Unfortunately there are serious drawbacks that need to be adressed before acceptance of the manuscript:

  1. Please consider that you need to submit your DNA data to Genbank and provide genbank accesion numbers.
  2. Please provide a tree to atteste bacterial identification
  3. you claimed in material section that :"The amplified products were sequenced (STAB VIDA, Caparica, Portugal) and the resulting sequences were compared in BLASTN gene bank. The percentage of identity accepted was greater than 90%." I am sorry but 90% does not even the genus level !!!!! please provide raw data of sequences.

Please adress these issues to provide relevant data to journal readers

Best regards

Author Response

Dear reviewer, thank you for all your comments and contributions to the improvement of our manuscript. 

Round 2

Reviewer 1 Report

Many suggestions were incorporated and the manuscript improved considerably. However, some comments were not properly addressed although the authors claimed in the rebuttal that they were; and the revised manuscript still has a room for improvement.

  1. Discussion still needs to be restructured and more streamlined, in particular, in lines 272-311.
  2. Lines 272-279: This paragraph does not fit into the flow of Discussion and I would advise to destructure this part into pieces and insert them into appropriate places in Discussion.
  3. The end of Conclusion is rather weak and needs to be reworked.
  4. Lines 219 and 220: This statement does not make sense. Why are their percentages 33 and 25% when they were found in all samples? Double-check the numbers and/or revise it. Cite "Table 2" at the end of the sentence.
  5. How many replicates were used for each antibiotic? Did five antibiotic discs per plate all contain different antibiotics?
  6. Italicize names of all the microorganisms properly throughout the manuscript.
  7. In figure legends, words are often capitalized in the middle of a sentence. Correct this throughout the manuscript.

Specific comments:

Simple summary

          Line 15: Change "was evaluated" to "were evaluated".

Abstract

          Lines 23 and 24: Clarify and revise "this has resulted ... use of antibiotics."

          Line 32: Change "E coli" to "E. coli".

Introduction

          Line 55: Change "water, sanitation," to "water-, sanitation-,"

          Lines 59 and 60: Provide references. This was mentioned in the previous review but was not addressed without proper justification.

          Lines 65-67: Provide references.

          Lines 75-81: This is redundant with lines 72-74 and I recommend merging these two parts. Also, "Consequently" in line 79 does not make sense; revise it.

Materials and Methods

          Line 89-90: Provide references.

          Lines 108-110: Add "n=" before the number of samples.

          Line 111: Change "and immediately placed" to "immediately placed".

          Line 122: Add a period after "standards [31]".

          Lines 122 and 123: Change "(Biokar 122 Diagnostics; Beauvais, France)" to "(Biokar 122 Diagnostics, Beauvais, France)". Also, change other similar instances where a company name is separated from its geographical location by a semi-colon.

          Line 130: Change "(Biokar Diagnostics; Beauvais, France)" to "(Biokar Diagnostics)" since its geographical location was already mentioned. Revise other instances appropriately.

          Line 144: Add a comma after "namely". This was mentioned in the previous review but was not addressed without proper justification.

          Line 153: Revise "in the BLASTN gene bank". Also, reference 36 is not appropriate. Refer to the BLAST reference website (https://blast.ncbi.nlm.nih.gov/Blast.cgi?CMD=Web&PAGE_TYPE=BlastDocs&DOC_TYPE=References).

          Line 159: In the current manuscript, "disc" ("discs") and "disk" are used. Use one term consistently.

          Line 179: Change "in the CLSI" to "by the CLSI".

Results

          Line 189: Change "Chromogenic Coliform Agar (CCA) plates" to "CCA plates". Once an acronym is defined, use the acronym consistently. Revise similar instances throughout the manuscript.

          Lines 189 and 190: Cite Table 2.

          Lines 190-192: Cite Table 2.

          Line 194: Change "four were from well water" to "four from well water". This was mentioned in the previous review but was not addressed without proper justification.

          Line 218: Change "12 (100%) water samples" to "12 water samples (100%)".

          Line 257: Change "However, in non-beta-lactams," to "For non-beta-lactams,".

          Line 262: Change "more frequently occurred" to "occurred more frequently".

Discussion

          Lines 281-283: Add references.

          Line 284: Re-write "However,"

          Line 291: Re-write "in some cases, often" since it sounds redundant.

          Lines 291 and 292: Change "In an initial approach" to "At the first glance".

          Lines 307 and 308: The statement "In addition, ... analyzed" pertains to tap water although this paragraph discusses water sources. I would recommend removing this statement. Also, be more specific about "analyzed"; change "analzyed" to "analzyed in this study".

          Lines 322 and 323: "In addition" and "also" are redundant. I would suggest removing "also".

          Lines 331 and 344: Add a period after "et al".

          Lines 348 and 349: Specify "intermediate susceptibility".

Table 1

          Change "Home bottled" to "Home-bottled".

Table 3

          Line 210: Remove "(average values ± standard deviation of log CFU/ml or log CFU/100 ml)" since it is shown in the legend. Revise similar instances in other tables.

          Line 211: Change "are indicated" to "is indicated". Revise similar instances in other tables and legends.

          Line 215: Revise "Fecal enterococci Fecal coliforms and E. coli" for clarity.

Table 6

          Line 247: Change "28 of E. coli and 16 of Klebsiella spp." to "28 isolates of E. coli and 16 isolates of Klebsiella spp.".

Table 7

          Line 264: Change "non susceptible" to "non-susceptible". This was mentioned in the previous review but was not addressed without proper justification.

Author Response

Answers to Reviewer 1.

Many suggestions were incorporated and the manuscript improved considerably. However, some comments were not properly addressed although the authors claimed in the rebuttal that they were; and the revised manuscript still has a room for improvement.

R: Dear reviewer, thank you once again for all your comments and contributions to the improvement of our manuscript. The highlighted of the answers were posted in “Track Changes”.

  1. Discussion still needs to be restructured and more streamlined, in particular, in lines 272-311.

Answer: Done. According to the comments in the discussion section

  1. Lines 272-279: This paragraph does not fit into the flow of Discussion and I would advise to destructure this part into pieces and insert them into appropriate places in Discussion.

Answer: Part of the text was reused for discussion, another part was removed

  1. The end of Conclusion is rather weak and needs to be reworked.

Answer: The end of the conclusion has been revised.

  1. Lines 219 and 220: This statement does not make sense. Why are their percentages 33 and 25% when they were found in all samples? Double-check the numbers and/or revise it. Cite "Table 2" at the end of the sentence.

Answer: This statement refers to contamination by mesophiles and Fecal enterococci that were found in 100% of the samples (twelve sample from wells analyzed) and to Fecal coliforms and E. coli that were present in more than 80% of the samples.

  1. How many replicates were used for each antibiotic? Did five antibiotic discs per plate all contain different antibiotics?

Answer: Two replicates were used for each antibiotic. Yes, were five discs of different antibiotics for each plate. Added information (lines: 170-171)

  1. Italicize names of all the microorganisms properly throughout the manuscript.

Answer: Done

  1. In figure legends, words are often capitalized in the middle of a sentence. Correct this throughout the manuscript.

Answer: Done

Specific comments:

Simple summary

  1. Line 15: Change "was evaluated" to "were evaluated".

 Answer: Done (line: 15)

Abstract

  1. Lines 23 and 24: Clarify and revise "this has resulted ... use of antibiotics."

 Answer: Done (lines: 23-24)

  1. Line 32: Change "E coli" to "E. coli".

Answer: Done (line: 32)

 Introduction

  1. Line 55: Change "water, sanitation," to "water-, sanitation-,"

 Answer: Done (line: 55)

  1. Lines 59 and 60: Provide references. This was mentioned in the previous review but was not addressed without proper justification.

Answer: Done (line: 61). (It was done previously but was changed at the time of the English edition. We apologize. Now it is correct)

  1. Lines 65-67: Provide references.

Answer: Done (line: 68)

  1. Lines 75-81: This is redundant with lines 72-74 and I recommend merging these two parts. Also, "Consequently" in line 79 does not make sense; revise it.

Answer: Changes were made to the text (Line: 76-86)

 Materials and Methods

  1. Line 89-90: Provide references.

Answer: Done (line: 93)

  1. Lines 108-110: Add "n=" before the number of samples.

Answer: Done (line: 110-112)

  1. Line 111: Change "and immediately placed" to "immediately placed".

Answer: Done (line: 113)

  1. Line 122: Add a period after "standards [31]".

Answer: Done (line: 123)

  1. Lines 122 and 123: Change "(Biokar 122 Diagnostics; Beauvais, France)" to "(Biokar 122 Diagnostics, Beauvais, France)". Also, change other similar instances where a company name is separated from its geographical location by a semi-colon.

Answer: Done in all

  1. Line 130: Change "(Biokar Diagnostics; Beauvais, France)" to "(Biokar Diagnostics)" since its geographical location was already mentioned. Revise other instances appropriately.

Answer: Done in all

  1. Line 144: Add a comma after "namely". This was mentioned in the previous review but was not addressed without proper justification.

Answer: Done (line: 146)

  1. Line 153: Revise "in the BLASTN gene bank". Also, reference 36 is not appropriate. Refer to the BLAST reference website (https://blast.ncbi.nlm.nih.gov/Blast.cgi?CMD=Web&PAGE_TYPE=BlastDocs&DOC_TYPE=References).

Answer: Done

  1. Line 159: In the current manuscript, "disc" ("discs") and "disk" are used. Use one term consistently.

Answer: Done in all text

  1. Line 179: Change "in the CLSI" to "by the CLSI".

Answer: Done (line: 182)

 Results

  1. Line 189: Change "Chromogenic Coliform Agar (CCA) plates" to "CCA plates". Once an acronym is defined, use the acronym consistently. Revise similar instances throughout the manuscript.

Answer: Done (line: 192)

  1. Lines 189 and 190: Cite Table 2.

Answer: Done (line: 193)

  1. Lines 190-192: Cite Table 2.

Answer: Done (line: 195)

  1. Line 194: Change "four were from well water" to "four from well water". This was mentioned in the previous review but was not addressed without proper justification.

Answer: Done (line: 198). (It was done previously but was changed at the time of the English edition. We apologize. Now it is correct)

  1. Line 218: Change "12 (100%) water samples" to "12 water samples (100%)".

Answer: Done (line: 222) (It was done previously but was changed at the time of the English edition. We apologize. Now it is correct)

  1. Line 257: Change "However, in non-beta-lactams," to "For non-beta-lactams,".

Answer: Done (line: 26)

  1. Line 262: Change "more frequently occurred" to "occurred more frequently".

Answer: Done (line: 266-267)

 Discussion

  1. Lines 281-283: Add references.

Answer: Done (line: 287-288)

  1. Line 284: Re-write "However,"

Answer: Done (line: 289)

  1. Line 291: Re-write "in some cases, often" since it sounds redundant.

Answer: Done (line: 295)

  1. Lines 291 and 292: Change "In an initial approach" to "At the first glance".

Answer: Done (line: 295-296)

  1. Lines 307 and 308: The statement "In addition, ... analyzed" pertains to tap water although this paragraph discusses water sources. I would recommend removing this statement. Also, be more specific about "analyzed"; change "analzyed" to "analzyed in this study".

Answer: Done. this statement was removed.

  1. Lines 322 and 323: "In addition" and "also" are redundant. I would suggest removing "also".

Answer: Done.  "also" was removed (327)

  1. Lines 331 and 344: Add a period after "et al".

Answer: Done (line: 255)

  1. Lines 348 and 349: Specify "intermediate susceptibility".

         Answer: Done (line: 352)

Table 1

  1. Change "Home bottled" to "Home-bottled".

Answer: Done

Table 3

  1. Line 210: Remove "(average values ± standard deviation of log CFU/ml or log CFU/100 ml)" since it is shown in the legend. Revise similar instances in other tables.

Answer: Done

  1. Line 211: Change "are indicated" to "is indicated". Revise similar instances in other tables and legends.

Answer: Done

  1. Line 215: Revise "Fecal enterococci Fecal coliforms and E. coli" for clarity.

Answer: Done. It was separated by comma.

Table 6

  1. Line 247: Change "28 of E. coli and 16 of Klebsiella spp." to "28 isolates of E. coli and 16 isolates of Klebsiella spp.".

Answer: Done.

Table 7

  1. Line 264: Change "non susceptible" to "non-susceptible". This was mentioned in the previous review but was not addressed without proper justification.

Answer: Done

Reviewer 2 Report

Dear authors,

All my recommendations have been fulfilled therefore I suggest acceptance of the manuscript.

Best regards

Author Response

Answers to the reviewer 2.

Dear authors,

All my recommendations have been fulfilled therefore I suggest acceptance of the manuscript.

Best regards

R: Thank you very much for your comments.
